# Direct conversion of methane to formaldehyde and CO on B$_2$O$_3$ catalysts

Jinshu Tian[1,3], Jiangqiao Tan[1,3], Zhaoxia Zhang[1,3], Peijie Han[1], Min Yin[1], Shaolong Wan[1], Jingdong Lin [1], Shuai Wang [1✉] & Yong Wang [2✉]

Direct oxidation of methane to value-added C$_1$ chemicals (e.g. HCHO and CO) provides a promising way to utilize natural gas sources under relatively mild conditions. Such conversions remain, however, a key selectivity challenge, resulting from the facile formation of undesired fully-oxidized CO$_2$. Here we show that B$_2$O$_3$-based catalysts are selective in the direct conversion of methane to HCHO and CO (~94% selectivity with a HCHO/CO ratio of ~1 at 6% conversion) and highly stable (over 100 hour time-on-stream operation) conducted in a fixed-bed reactor (550 °C, 100 kPa, space velocity 4650 mL g$_{cat}^{-1}$ h$^{-1}$). Combined catalyst characterization, kinetic studies, and isotopic labeling experiments unveil that molecular O$_2$ bonded to tri-coordinated BO$_3$ centers on B$_2$O$_3$ surfaces acts as a judicious oxidant for methane activation with mitigated CO$_2$ formation, even at high O$_2$/CH$_4$ ratios of the feed. These findings shed light on the great potential of designing innovative catalytic processes for the direct conversion of alkanes to fuels/chemicals.

[1] State Key Laboratory for Physical Chemistry of Solid Surfaces, Collaborative Innovation Center of Chemistry for Energy Materials, National Engineering Laboratory for Green Chemical Productions of Alcohols-Ethers-Esters, and College of Chemistry and Chemical Engineering, Xiamen University, 361005 Xiamen, China. [2] Voiland School of Chemical Engineering and Bioengineering, Washington State University, Pullman, WA 99164, USA. [3]These authors contributed equally: Jinshu Tian, Jiangqiao Tan, Zhaoxia Zhang. ✉email: shuaiwang@xmu.edu.cn; yongwang@pnnl.gov

The catalytic transformation of methane to value-added chemicals is of significant interest for the efficient utilization of natural gas sources[1,2], especially due to the recent shale-gas revolution[3]. In particular, the direct conversion of methane via selective oxidation to $C_1$ chemicals (i.e. $CH_3OH$, HCHO, and CO) is most attractive, because all of these products are important platform molecules/intermediates for the production of fuels and chemicals[4,5]. Selective oxidation of methane involves the cleavage of C–H bonds and formation of C–O bonds, leading to $CH_3OH$, HCHO, and CO as the desired partial oxidation products but also to $CO_2$ as an undesired complete oxidation product. Such oxidation processes are induced by nucleophilic attack of active O species at the H or C atom in $CH_4$ (or reactive intermediates) before electron transfer[6]. Because the desired $C_1$ partial oxidation products (i.e. $CH_3OH$, HCHO, and CO) have much higher electrophilicity than $CH_4$, they are kinetically more favored in oxidation, leading to dominant formation of undesired $CO_2$ at methane conversions even less than 5% on conventional catalysts[7–12] (e.g., $V_2O_5$[13], $MoO_3$[14], and $Fe_2O_3$[15]). Lattice O anions exposed on these oxide surfaces act as the nucleophilic and oxidative centers that can be regenerated fast via dissociative adsorption of gaseous $O_2$ during catalytic cycles (also described as the Mars van Krevelen mechanism[16]). To our best knowledge, high selectivities to the desired $C_1$ partial oxidation products on these traditional metal oxide catalysts are able be obtained merely at low methane conversions (<2%) or low $O_2/CH_4$ ratios (<0.5)[7–15], making such oxidation processes impractical.

Directly using molecular $O_2$ or derived O· radicals to oxidize $CH_4$, instead of the more nucleophilic and reactive lattice O anions, would render facile control over the extent of $CH_4$ oxidation. Recent studies[17,18] have shown that nonmetallic B-based materials (e.g. BN[19–21], $B_4C$[22], $SiB_6$[23], and $B_2O_3$[24,25]) can catalyze oxidative dehydrogenation of $C_2$–$C_4$ alkanes to alkenes with extraordinarily low selectivities to $CO_2$, reflecting, in turn, that these catalysts probably use the moderate $O_2$ or O· oxidants to activate the alkane reactants. The BN catalyst has also been attempted for methane oxidation at temperatures above 690 °C, which yielded CO, $CO_2$, and methane coupling products (i.e. $C_2H_4$ and $C_2H_6$) with respective selectivities of 76.6, 4.3, and 19.3 at 20.5% methane conversion[26]. We expect that the selectivity of valuable $C_1$ products for methane oxidation on the B-based catalysts can be significantly improved at milder temperatures under which HCHO and $CH_3OH$ can be better stabilized kinetically. Moreover, experiments of nuclear magnetic resonance spectroscopy[27] and X-ray photoelectron spectroscopy[22] have revealed that $B(OH)_xO_{3-x}$ (where $x = 0$–3) layers formed on the B-based catalysts act as the true active phase, irrespective of their bulk contents.

According to the above considerations, we here study supported $B_2O_3$ catalysts for selective methane oxidation at relatively mild temperatures. Our results show that $B_2O_3$-based catalysts are highly selective in the direct conversion of methane to HCHO and CO, and these selectivities are unexpectedly insensitive to the $O_2/CH_4$ ratios. Structural characterization, kinetic measurements, and isotopic labeling experiments are combined to discern that molecular $O_2$ bonded to coordinately unsaturated $BO_3$ centers on the $B_2O_3$ surfaces is the crucial oxidant that accounts for the selective methane oxidation. The mechanistic understanding of methane oxidation on the unique $B_2O_3$ surfaces would inspire the design of the next generation of heterogeneous catalysts for selective oxidation of hydrocarbons.

## Results

**Performances of $B_2O_3$-based catalysts in methane oxidation.** Catalysts containing 20 wt% $B_2O_3$ supported on various oxides (i.e. $Al_2O_3$, $SiO_2$, ZnO, $TiO_2$, and $ZrO_2$) were prepared using the wetness impregnation method with boric acid ($H_3BO_3$) as the boron source (see Methods section). This high $B_2O_3$ loading was chosen to ensure that multiple $B_2O_3$ layers were formed on each oxide support (Supplementary Table 1). Under the conditions studied ($O_2/CH_4$ ratio of 1.0 and 550 °C), these oxide supports themselves were inactive for methane activation, whereas all supported catalysts with 20 wt% $B_2O_3$ selectively converted methane to HCHO and CO (~94 % selectivity with a HCHO/CO ratio of ~1), together with a trace amount of desired $CH_3OH$ and $C_2$ products ($C_2H_4$ and $C_2H_6$), which are irrespective of the nature of the support (Fig. 1a). It is noteworthy that these observed conversions are mainly attributed to the catalytic processes on the $B_2O_3$ surface, instead of gas-phase radical reactions, because the conversion and selectivity of methane oxidation on the supported $B_2O_3$ catalysts had negligible changes whether the empty space of the reactor was fully filled with inert SiC material or not (Supplementary Fig. 1) and their sum did not obey the

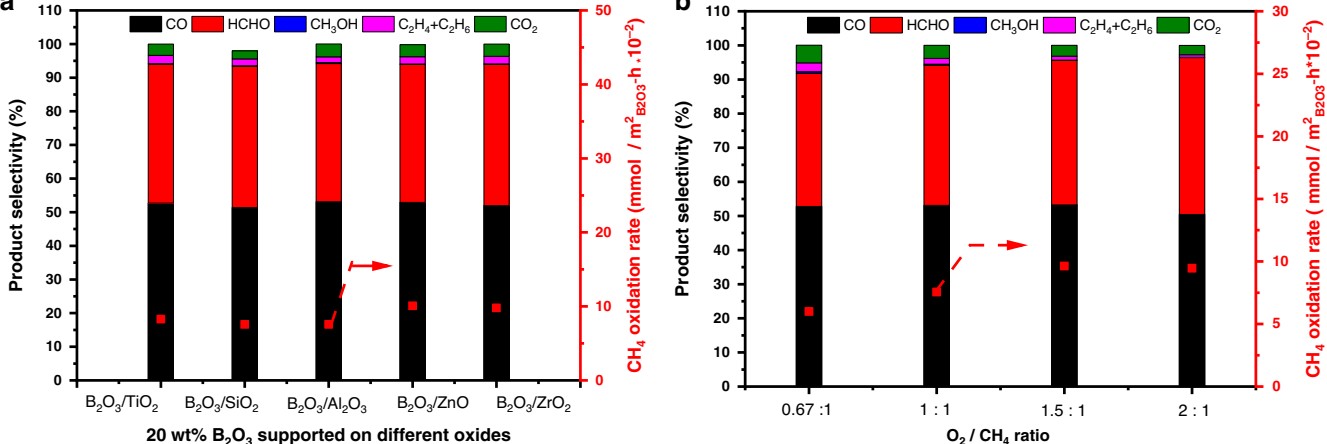

**Fig. 1 Methane oxidation rates and selectivity on supported $B_2O_3$ catalysts. a** Effects of oxide support on 20 wt% $B_2O_3$-based catalysts. **b** Effects of $O_2/CH_4$ ratio on 20 wt% $B_2O_3/Al_2O_3$. Reaction conditions: 550 °C, 32 kPa $P_{CH4}$, 32 kPa $P_{O2}$ for (**a**) or 21–64 kPa $P_{O2}$ for (**b**), gas composition balanced with $N_2$, ~6% $CH_4$ conversion was achieved by adjusting the space velocity within a range of 1500–50000 mL $g_{cat.}^{-1}$ $h^{-1}$. The methane oxidation rates reported here were normalized by the exposed surface area of $B_2O_3$. The bars in **a** and **b** denote product selectivities (CO in black, HCHO in red, $CH_3OH$ in blue, $C_2H_4$ and $C_2H_6$ in magenta, and $CO_2$ in green), and the red square in **a** and **b** denote methane oxidation rates.

empirical 100% rule[28] that was observed in the previous gas phase chemistry (Supplementary Fig. 2).

In all cases, the sum of the selectivities to these desired $C_1$ and $C_2$ products were above 96%, while the selectivity to undesired $CO_2$ was below 4% at ~6% methane conversion (controlled by the space velocity for rigorous selectivity comparison among the examined catalysts). Such high selectivities to the partial oxidation products reflect the superior control of the oxidation extent within the kinetic regime, brought forth by the unique property of the $B_2O_3$-based catalysts as described below, otherwise the fully oxidized $CO_2$ would be predominant among the products if the thermodynamic equilibrium is established (e.g., 53.5% methane conversion and 72.8% $CO_2$ selectivity at 550 °C and 100 kPa with an initial $CH_4/O_2$ molar ratio of 1/1; Supplementary Fig. 3). This highly selective methane oxidation process could be scaled up by combining efficient separation of the products from the effluent and recycling of the unconverted $CH_4$ and $O_2$ reactants[29]. Moreover, the fact that the molar ratio of HCHO to CO in the products is ~1.0 makes them potentially desired for the downstream acetic acid synthesis via hydrogenation of HCHO to methanol and its subsequent carbonylation[30,31] and also for glycolic acid synthesis via direct carbonylation of HCHO[32]. The detected $C_2H_6$ and $C_2H_4$ products are likely ascribed to methane oxidative coupling as we reported previously[21,26], reflecting the ability of $B_2O_3$-based catalysts to produce $C_{2+}$ molecules from direct methane oxidation. Similar methane oxidation rates (mmol methane converted per surface area of the exposed $B_2O_3$ phase per hour) were obtained for the examined catalysts. These results suggest a negligible effect of the oxide supports on catalytic performance, a result that is consistent with formation of the multiple $B_2O_3$ layers on each oxide support at 20 wt% $B_2O_3$ loading (the loading threshold for forming a $B_2O_3$ monolayer for each of the oxide supports are shown in Supplementary Table 1).

We further studied the effects of $O_2/CH_4$ partial pressure ratios on the performances of the 20 wt% $B_2O_3/Al_2O_3$ catalyst at 550 °C, where methane conversions were kept ~6% via adjusting the space velocity for rigorous comparison of reactivity and selectivity. As shown in Fig. 1b, nearly constant selectivities (i.e., >95% selectivity to desired $C_1$ and $C_2$ products with <5% selectivity to undesired $CO_2$) were obtained at ~6% methane conversion over a wide range of $P_{O2}/P_{CH4}$ ratios (0.67–2.00), reflective of the unique ability of $B_2O_3$ in preventing complete oxidation of the desired $C_1$ and $C_2$ products to $CO_2$ and thus the potential of high-pressure operation for this catalytic process. Such remarkable selectivities to the desired $C_1$ and $C_2$ products and a negligible effect of the $P_{O2}/P_{CH4}$ ratio on activity makes this system superior to conventional oxide catalysts (e.g. supported $V_2O_5$ and $MoO_3$ in Supplementary Table 2), which must operate at much lower $P_{O2}/P_{CH4}$ ratios (0.1–0.5) to minimize the overoxidation of $C_1$ and $C_2$ products by the strongly nucleophilic lattice O anions on metal oxide surfaces[16,33]. Figure 1b also shows that methane oxidation rates increased with increasing the $O_2/CH_4$ ratio, indicating a positive reaction order with respect to $O_2$. In addition, these $B_2O_3$-based catalysts exhibited exceptional stability at 550 °C as illustrated over 100 h time-on-stream stable methane oxidation for the $B_2O_3/Al_2O_3$ catalyst (Supplementary Fig. 4).

**Active sites of $B_2O_3$-based catalysts for methane oxidation**. To provide insight into the nature of active sites on these supported $B_2O_3$ catalysts, [11]B solid state nuclear magnetic resonance (NMR) was used. Two boron oxide species with chemical shifts centered at 67.9 and 57.1 ppm were observed for these samples (using $NaBH_4$ as the reference compound, NMR spectra shown in

Supplementary Fig. 5). These are ascribed to tri-coordinated $BO_3$ and tetra-coordinated $BO_4$ units, respectively[34]. Previous studies[35,36] suggest that the $BO_4$ units are derived from the additional bonding of lattice O anions of the oxide support to the $BO_3$ units in $B_2O_3$. The $BO_4/BO_3$ molar ratio of the supported $B_2O_3$ catalysts increased with increasing basic strength of the oxide support (e.g. 0.46 for $B_2O_3/SiO_2$, 1.45 for $B_2O_3/Al_2O_3$, and 1.65 for $B_2O_3/ZrO_2$ in Supplementary Fig. 5) that can be explained by increased coordination of the B center to more basic lattice O anions of the oxide support. On the other hand, these $BO_4/BO_3$ ratios did not affect the methane oxidation rates of the supported $B_2O_3$ catalysts that were almost identical when normalized by the exposed $B_2O_3$ surface area (Fig. 1a). These results might indicate that the $BO_3$ and $BO_4$ units have similar catalytic activities.

To further confirm or disapprove our hypothesis, B-substituted ZSM-5 zeolite (B-ZSM-5) was prepared, where the B atoms embedded in the silicate framework were all tetra-coordinated by lattice O anions as confirmed by the single chemical shift of B at 54.1 ppm (Supplementary Fig. S5)[37,38]. Compared with the supported $B_2O_3$ catalysts (e.g., 20 wt% $B_2O_3/Al_2O_3$), the B-ZSM-5 sample showed negligible activity in catalyzing methane oxidation with near 45% selectivity to $CO_2$ (Supplementary Fig. 6). It is thus concluded that the $BO_4$ units with full coordination of B centers are catalytically inactive. The fact that the methane oxidation rates of the supported $B_2O_3$ catalysts are independent of the $BO_4/BO_3$ ratio measured by [11]B NMR is likely due to that NMR is a bulk technique and that the top layer $B_2O_3$ is not bonded to the oxygen anions of oxide supports on these catalysts with multilayered $B_2O_3$, leading to exclusively active surface $BO_3$ units for methane oxidation.

**Mechanism of methane oxidation on $B_2O_3$-based catalysts**. Kinetic studies were carried out to provide molecular-level insights into the mechanism of methane activation on supported $B_2O_3$ catalysts. The conversion-selectivity relationship for methane oxidation on 20 wt% $B_2O_3/Al_2O_3$ (Supplementary Fig. 7) showed that the selectivities to HCHO and $CH_3OH$ both monotonically decreased as the conversion of $CH_4$ increased from 2 to 10%, concomitant with increased selectivities to CO and $CO_2$. These selectivity trends indicate that both HCHO and $CH_3OH$ are primary products formed from methane conversion. It has been proposed that the initial activation of methane by active O species on oxide surfaces forms bonded methoxy intermediates ($CH_3O$) via cleaving a C–H bond in methane, which undergo further dehydrogenation to form HCHO or recombine the cleaved H atom to form $CH_3OH$[39,40]. On $B_2O_3$ surfaces, methane selective oxidation to HCHO is the predominant reaction channel for methane activation, as evidenced by the high $HCHO/CH_3OH$ selectivity ratios obtained at low methane conversions (e.g. ~17 at 2% conversion, Supplementary Fig. 7). The secondary dehydrogenation of HCHO leads to the formation of CO, whereas the $CO_2$ product comes from CO oxidation.

Figure 2a depicts that the methane oxidation rate increased with $O_2$ pressure (0–80 kPa; 550 °C), but the dependence became weaker at higher $O_2$ pressure that is indicative of higher $O_2$ coverages on the $B_2O_3$ surface. In contrast, the methane oxidation rate increased linearly with the $CH_4$ pressure in the same pressure range, suggesting that $CH_4$ barely adsorbs on the $BO_3$ sites during catalysis (Fig. 2b), which is consistent with the unfavorable adsorption of $CH_4$ on solid surfaces brought forth by its highly symmetrical geometry and weakly polarized C–H bonds[11]. These effects of reactant concentrations on the oxidation rates are indicative of the Eley-Rideal mechanism[41] on catalytic surfaces, in

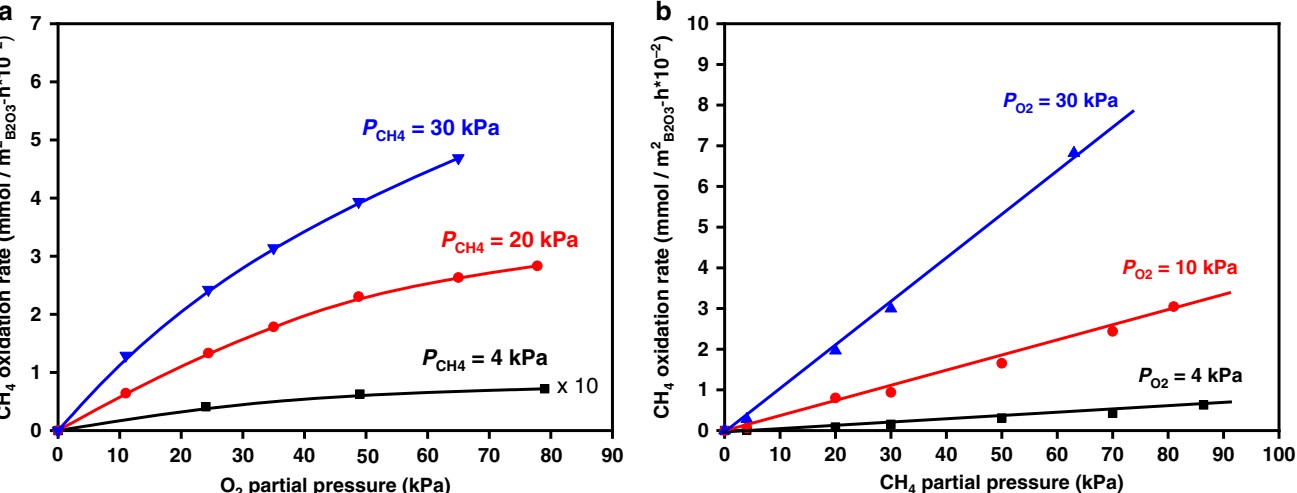

**Fig. 2 Kinetics of methane oxidation on supported $B_2O_3$ catalysts.** Methane oxidation rates as functions of **a** $O_2$ pressure and **b** $CH_4$ pressure were measured on 20 wt% $B_2O_3/Al_2O_3$. Reaction condition: 550 °C, 4–30 kPa $P_{CH_4}$, 0–80 kPa $P_{O_2}$ for **a** and 4–30 kPa $P_{O_2}$, 0–90 kPa $P_{CH_4}$ for **b**, balanced by $N_2$, space velocity at 4920 mL $g_{cat.}^{-1}$ $h^{-1}$. In **a**, the rate data measured at $CH_4$ partial pressures of 4, 20, and 30 kPa are shown as black square, red cycle, and blue triangle, respectively. In **b**, the rates measured at $O_2$ partial pressures of 4, 10, and 30 kPa were shown in black square, red cycle, and blue triangle, respectively. The curves in **a** and lines in **b** represent trends.

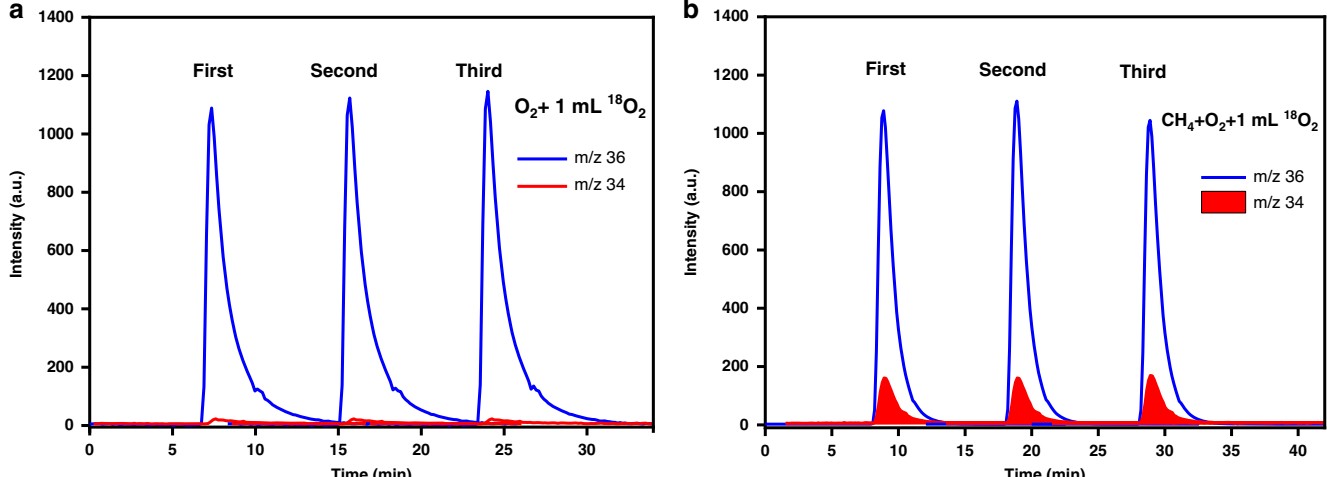

**Fig. 3 Isotopic assessment of $O_2$ activation on supported $B_2O_3$ catalysts.** Mass spectra of $^{16}O^{18}O$ ($m/z = 34$) species upon pulsing $^{18}O_2$ ($m/z = 36$) into $^{16}O_2$ flows on 20 wt% $B_2O_3/Al_2O_3$ were collected in the **a** absence and **b** presence of $CH_4$ ($CH_4$ 1 mL/min, $O_2$ 5 mL/min, $N_2$ 4 mL/min, temperature: 550 °C, catalyst loading: 0.2 g). In both **a** and **b**, the red and blue curves denote the signals for the $m/z$ ratios of 34 and 36, respectively.

which one molecule adsorbed on the active site directly reacts with another one from the gas phase. Because of this, we propose that methane oxidation occurs between a gaseous $CH_4$ molecule and an $O_2$ molecule bonded to the $BO_3$ sites. Similar Eley-Rideal-type pathways have been found for methane oxidation on Pd and Pt surfaces at high temperatures[42,43], which unveils that two adsorbed O atoms formed from the dissociation of a $O_2$ molecule on the catalyst surface are required to act concertedly in order to cleave the strong C–H bond of $CH_4$.

Isotopic labeling experiments were further used to confirm the above hypothesis (i.e. $BO_3$-surface-bonded $O_2$ directly activates $CH_4$). Pulses of a small amount of $^{18}O_2$ into flowing $^{16}O_2$ on the supported $B_2O_3$ catalysts at 550 °C did not lead to the formation of isotope-exchanged $^{18}O^{16}O$ species, excluding the presence of dissociative adsorption of $O_2$ molecules on the $B_2O_3$ surface (Fig. 3a). In contrast, when $CH_4$ was co-fed with $^{16}O_2$ and pulses of $^{18}O_2$ over the surface of the same $B_2O_3$ catalysts, a significant amount of $^{18}O^{16}O$ was detected (Fig. 3b). We thus infer that the

O–O bond in the adsorbed $O_2$ molecule is cleaved concertedly when it activates the C–H bond of $CH_4$.

The above kinetic and isotopic assessments are combined to give a plausible pathway for the formation of HCHO from $CH_4$ oxidation on $B_2O_3$-based catalysts as described in Fig. 4a. First, $O_2$ adsorbs on two vicinal $BO_3$ units with each O atom in $O_2$ bound to one of the electron-deficient B centers (Step 1, Fig. 4a). A gaseous $CH_4$ molecule then attacks this adsorbed $O_2$ reactant, resulting in concurrent formations of hydroxy and methoxy species (Step 2, Fig. 4a). Formaldehyde and $H_2O$ are further produced from hydrogen abstraction of the methoxy moiety by a hydroxyl (Step 3, Fig. 4a). Desorption of these products from the active $BO_3$ sites completes a catalytic turnover for methane partial oxidation on $B_2O_3$ (Step 4, Fig. 3a). The measurable O-exchange between $O_2$ molecules in the presence of $CH_4$ (Fig. 3b) indicates the reversibility of Steps 1 and 2 on $B_2O_3$ under reaction conditions. This suggests, in turn, that Step 3 is kinetically relevant. This finding is consistent with a previous report[6] that

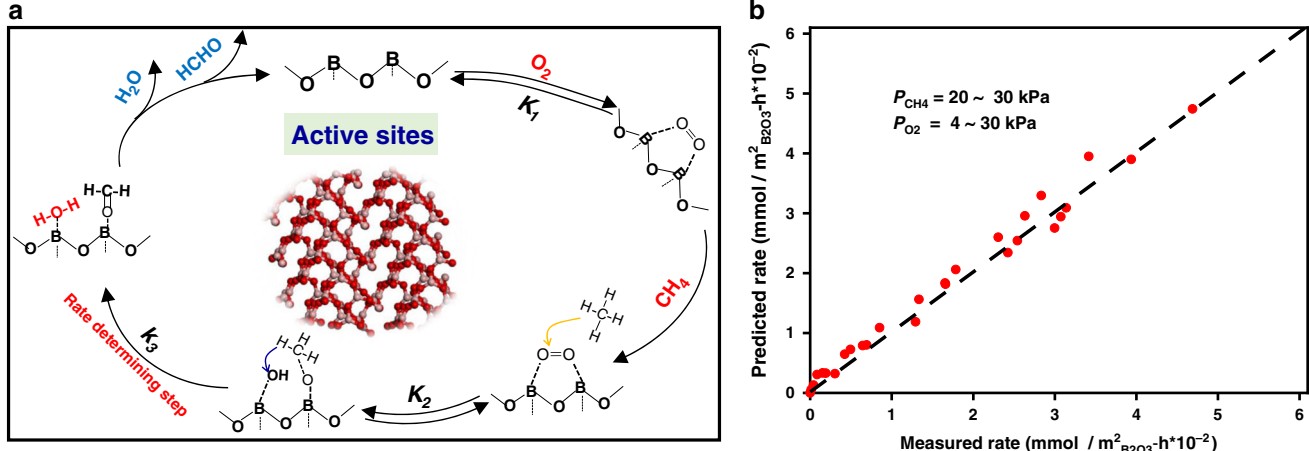

**Fig. 4 Mechanistic insights for methane activation on a B$_2$O$_3$ surface. a** Schematic diagram of the plausible pathway of methane selective oxidation to formaldehyde on B$_2$O$_3$-based catalysts. **b** Parity plots for the measured rate data of methane selective oxidation on B$_2$O$_3$/Al$_2$O$_3$ and those predicted using Eq. 1 (the regression-fitted parameters shown in Supplementary Table 3). In **a**, $K_1$ and $K_2$ are equilibrium constants for the corresponding steps, and $k_3$ is the kinetic constant for hydrogen abstraction of the surface methoxy species by a neighboring hydroxyl.

the electronegativity of oxide catalysts affects the selectivity to HCHO. These elementary steps, taken together with the pseudo-steady-state approximation for all bound species and the quasi-equilibrated nature of all steps except Step 3, lead to an equation for methane conversion rates ($r$):

$$r = \frac{k_3 K_1 K_2 P_{O2} P_{CH4}}{1 + K_1 P_{O2}} \quad (1)$$

Here, $K_1$ and $K_2$ are the respective equilibrium constants for Steps 1 and 2, whereas $k_3$ is the kinetic constant for Step 3. The functional form of Equation 1 accurately describes all methane oxidation rate data measured within a wide reactant pressure range (Fig. 4b; regression-fitted parameters shown in Supplementary Table 3), supporting the proposed methane activation mechanism on the B$_2$O$_3$ surfaces (Fig. 4a).

## Discussion

In summary, nonmetallic B$_2$O$_3$-based catalysts are selective and stable in the partial oxidation of methane to HCHO and CO. Surface tri-coordinated BO$_3$ units are the active sites for methane oxidation. O$_2$ molecules bound to the electron-deficient B centers of these BO$_3$ units are moderate oxidants for methane activation, exhibiting strong suppression of the formation of thermodynamically favored CO$_2$. Further exploitation of such nonmetallic oxide catalysts will bring innovative strategies and catalyst systems for efficient and selective oxidation of methane (and other alkanes) to valuable chemicals.

## Methods

**Preparation of B$_2$O$_3$-based catalysts**. B$_2$O$_3$ catalysts supported on various oxides (including Al$_2$O$_3$, TiO$_2$, ZnO, ZrO$_2$, and SiO$_2$) were prepared by wetness impregnation method using boric acid (Sinopharm chemical reagent co. LTD) as the boron source. The preparation method was described below using B$_2$O$_3$/Al$_2$O$_3$ as an illustrative example: a certain amount of H$_3$BO$_3$ was dissolved in deionized water (10 mL) and then the resulting aqueous solution was added dropwise dropped to pure Al$_2$O$_3$ (0.5 g). After stirring vigorously for 1 h, the impregnated samples were heated to 65 °C and subsequently vacuumed at 65 °C for 8 h. The as-prepared products were then calcined at 600 °C for 5 h in air.

**Preparation of B-ZSM-5 samples**. B-ZSM-5 was synthesized according to a previously reported method[37]. Typically, NaOH, 1–6 hexanediamine (HMDA) and tetrapropylammonium bromide (TPABr) were dissolved in 18.9 mL of deionized water. The solution was stirred for 30 min, and then 1.8 g of porous SiO$_2$ (Aladdin) and 0.3373 g H$_3$BO$_3$ were slowly added. The gel was stirred for 1 h and then transferred into a teflon-lined stainless-steel autoclave for crystallization

at 180 °C for 48 h. The obtained samples were separated by filtration, washed with deionized water, dried at 90 °C for 12 h and finally calcined under static air at 550 °C for 5 h.

**Measurement of catalytic performance**. Catalytic methane oxidation was conducted using a fixed-bed quartz tubular reactor (7 mm inner dimeter) with plug-flow hydrodynamics. The B$_2$O$_3$-based catalyst (0.15–0.18 mm sieved particles, ~200 mg, corresponding to a volume of 0.38 mL) was first pretreated in flowing O$_2$/N$_2$ (1/1 in volume) for 2 h under 580 °C and then cooled to the reaction temperature under N$_2$. CH$_4$/He (90/10%), O$_2$ (99.99%), and N$_2$ (99.99%) were individually controlled using three mass flow controllers (Sevenstar Technology Co., Ltd) to provide the reaction gas feed. The feed rate reported here is the weight hourly space velocity (WHSV), in which the gas volume refers to the standard ambient temperature and pressure. The concentrations of reaction products in the effluent were analyzed by an online gas chromatography (GC2060, Shanghai Ruimin GC Instruments, Inc). Samples in the quantitative ring were separated by Porapak column (6 m×3 mm) and then quantified using a thermal conductivity detector (TCD) for He, CH$_4$, and CO$_2$[44]. The other gases are introduced into a flame ionization detector (FID) and subsequently analyzed including CO, CH$_4$, CO$_2$, C$_2$H$_4$, C$_2$H$_6$, HCHO, and CH$_3$OH. Control experiments with SiC showed that there was negligible methane conversion without the catalyst. In all tests, carbon mass balances exceeded 98%. The CH$_4$ conversion ($X_{CH4}$) and the carbon selectivity of each product $i$ ($S_i$) were calculated using a standard normalization method (He as internal standard gas) based on the carbon balance, which were defined as

$$X_{CH4} = \left(1 - \frac{P_{CH4}^{out}}{P_{CH4}^{in}}\right) \times 100\% \quad (2)$$

$$S_i = \frac{n_i P_i^{out}}{\sum n_i P_i^{out}} \times 100\% \quad (3)$$

Here, $P_{CH4}^{out}$ and $P_{CH4}^{in}$ are the corresponding partial pressures of methane at the outlet and inlet of the reactor, while $P_i^{out}$ and $n_i$ are the outlet partial pressure and the carbon number of each product $i$ formed from methane oxidation, respectively.

**Structural characterization**. Specific surface areas of catalysts were measured by the Brunauer–Emmett–Teller (BET) method, using a Micromeritics Tristar 3020 surface area and porosimetry analyzer. Prior to measurement, all samples were degassed at 150 °C for 6 h. $^{11}$B solid nuclear magnetic resonance ($^{11}$B-NMR) analysis was recorded on a Bruker NMR 500 DRX spectrometer at 500 MHz and referenced to NaBH$_4$ (3.2 ppm). Isotopic labeling experiments were performed in a fixed-bed single-pass flow micro-reactor. A mixture of N$_2$ and $^{16}$O$_2$ (research grade, 99.99%) was fed to the 20 wt% B$_2$O$_3$/Al$_2$O$_3$ catalyst bed at 550 °C until the baseline was stabilized and then an $^{18}$O$_2$ (Cambridge Isotope Lab., 99%; 1 mL each time) pulse was injected into the flow using a syringe. The chemical and isotopic compositions of the reactor effluent were measured by online mass spectrometry (MS, Pfeiffer, OminStar TM) at intervals of 10 s with a m/z scanning from 1 to 50. The m/z signals of 32, 34, and 36 represent $^{16}$O$^{16}$O, $^{16}$O$^{18}$O, and $^{18}$O$^{18}$O, respectively.

## Data availability

The datasets generated and analyzed during the current study are available from the corresponding authors upon a reasonable request.

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

## Acknowledgements

This work was supported by the National Natural Science Foundation of China (No. 21922201, 21872113, 91945301, 21673189, and 91545114) and the Fundamental Research Funds for the Central Universities (No. 20720190036 and 20720160032).

## Author contributions

Shu.W. and Y.W. conceived the idea for the project. J.Ti., J.Ta., and M.Y. conducted the material synthesis. P.H., M.Y., and J.Ta. performed the structural characterizations and catalytic test. Sha.W., Y.W., J.L., and Z.Z. discussed the catalytic results. J.Ti. drafted the manuscript under the guidance of Y.W. and Shu.W. All authors discussed and commented on the manuscript.

## Competing interests

The authors declare no competing interests.
