## [Peer Review File · Nature Communications]

Editorial Note: This manuscript has been previously reviewed at another journal that is not operating a transparent peer review scheme. This document only contains reviewer comments and rebuttal letters for versions considered at *Nature Communications* .

REVIEWER COMMENTS

Reviewer #1 (Remarks to the Author):

I have read carefully the response to all the referees' comments. The response to my comment that the reaction is a surface initiated gas phase reaction is that they have used SiC to fill the voids in the bed. I am not sure this really negates a gas phase effect and maybe they need to show this with different particle sizes of SiC. Also I presume SiC is inert (there were reports in the 1980s that it was). However, if the other referees are content then I do not want to stand in the way of publication as the authors do show interesting data. It's just the mechanism that is very different from what others may have found.

There are two further points. The authors show that the 100% rule may not fit but as conversion changes so the C balance may be affected and they should show the Carbon balance for their data.

Also there was a paper in Nature 30 years ago (J.S.J. Hargreaves, et al 'Control of product selectivity in the partial oxidation of methane', Nature, 348 (1990) 428-429.) which shows that the O₂ conversion is the controlling factor in observing formaldehyde or methanol or CO. Perhaps they can add the details on O₂ conversion with their work and does this fit with this earlier model.

Reviewer #2 (Remarks to the Author):

The authors present their work on the conversion of methane-oxygen mixtures to formaldehyde and CO over B₂O₃-based catalyst at temperatures of 550°C and nearly atmospheric pressure. The catalyst is tested in a fixed bed reactor, the product stream analyzed by GC. High selectivity to HCHO and CO (ratio 1) at rather low methane conversion is achieved. Aside from experimental data, a mechanistic scheme is proposed.

My major concern with this paper is the first sentence in the conclusion: "nonmetallic B₂O₃-based catalysts show superior selectivity and stability in selectively catalyzing the partial oxidation of methane to HCHO and CO." Superior to what? Concerning CO, there are several catalysts that exhibit much higher selectivity and (even more) yields than the one proposed, just think of all the syngas production processes. Similar for HCHO, there are well-established commercial processes with excellent performance available. Since the paper as well as the journal focuses on catalysts, the authors should at least give a turn-over-frequency of their catalyst and a comparison to the established ones.

Response to Reviewers

The point-by-point response to the two reviewers is addressed below. We appreciate greatly the reviewers' time and effort for improving this manuscript and have tried our best to revise it accordingly.

Response to Reviewer 1:

Reviewer 1: *I have read carefully the response to all the referees' comments. The response to my comment that the reaction is a surface initiated gas phase reaction is that they have used SiC to fill the voids in the bed. I am not sure this really negates a gas phase effect and maybe they need to show this with different particle sizes of SiC. Also, I presume SiC is inert (there were reports in the 1980s that it was). However, if the other referees are content than I do not want to stand in the way of publication as the authors do show interesting data. It is just the mechanism that is very the disagreement with others maybe found.*

Response: Gas-phase radical reactions are sensitive to the empty space of the fixed-bed reactor as shown by a recent report from Hermans's group at University of Wisconsin-Madison (*Org. Process Res. Dev.* 2018, 22, 1644–1652), in which they examined the influence of reactor parameters on the BN-catalyzed oxidative dehydrogenation of propane and used SiC as an inert catalyst diluent. Our ongoing study also shows that, distinct from those B₂O₃-based catalysts, methane oxidation on BN is governed by gas-phase chemistry, evidenced by: 1) the measured reaction order of methane is higher than one (Fig. R1, unpublished data; in contrast to the first order found for B₂O₃/Al₂O₃ as illustrated in Fig. 2b of the main text); and 2) there is a strong inhibition effect by filling the empty space with SiC on methane oxidation over BN as opposed to the case over B₂O₃ (Fig. R2, unpublished data). The negligible activity observed on BN with SiC filler also confirms SiC is inert as the reviewer presumed. These latest results reflect the removal of empty space in the reactor by SiC filling is an effective way to assess the contribution of gas-phase chemistry in methane oxidation, and confirm, in turn, that methane oxidation reaction on the supported B₂O₃ catalysts predominantly occurred on the catalyst surface, instead of in gas phase. This unique mechanistic feature renders the B₂O₃-based catalysts are much more selective to formaldehyde and CO in partial oxidation of methane than those reducible metal oxide catalysts (e.g. V₂O₅ and MoO₃ catalysts) that have been intensively studied in literature.

Figure R1. Methane oxidation rates as functions of CH₄ pressure on a commercial BN catalyst. Reaction condition: 550°C, 32 kPa P_{O₂}, 0-80 kPa P_{CH₄}, balanced by N₂, space velocity at 4920 mL g_{cat}⁻¹ h⁻¹, methane conversion varied within 1.4%~16%.

Figure R2. Comparison of the effects of SiC filling on methane oxidation reaction between 20 wt% B₂O₃/Al₂O₃ and commercial BN catalysts. Reaction conditions: 550°C, 32 kPa P_{CH₄}, 32 kPa P_{O₂}, balanced with N₂, inert SiC (0.18-0.25 mm) was used to fill the empty space of the fixed-bed reactor.

Reviewer 1: There are two further point. The authors show that the 100% rule may not fit but as conversion changes so the C balance may be affected and they should show the Carbon balance for their data.

Response: Thank you for this suggestion. We have double checked the carbon balance. As shown in Figure R3, the carbon balance was kept above 98% within the examined methane conversion range (4%-26%), which further confirms that methane oxidation on the supported B_2O_3 catalysts did not obey the empirical 100% rule observed in the previous gas phase chemistry. Figure R3 has been included in the supporting information as Figure S2. The relevant sentences have been added in Lines 10-11 of Page 4 as:

“..... and their sum did not obey the empirical 100% rule²⁸ that was observed in the previous gas phase chemistry (Fig. S2).”

Figure R3. The sum of the CH_4 conversion and selectivities to the partial oxidation products and the corresponding carbon balance as a function of CH_4 conversion for the 20 wt% B_2O_3/Al_2O_3 catalyst. Reaction conditions: 550°C, 32 kPa P_{CH_4} , 32 kPa P_{O_2} , gas composition balanced with N_2 , the CH_4 conversion was varied by adjusting the space velocity within a range of 4000-50000 mL $g_{cat}^{-1} h^{-1}$.

Reviewer 1: Also there was a paper in Nature 30 years ago (J.S.J. Hargreaves, et al 'Control of product selectivity in the partial oxidation of methane', Nature, 348 (1990) 428-429.) which shows that the O_2 conversion is the controlling factor in observing formaldehyde or methanol or CO. Perhaps they can add the details on O_2 conversion with their work and does this fit with this earlier model.

Response: Thank you for bringing this inspiring paper to our attention. This reference reports that the product selectivity of methane oxidation on undoped MgO catalysts at 1023 K shifted from formaldehyde to ethane and ethene as the conversion of O₂ increased, because the gas-phase coupling of methyl radical prevailed over the methyl radical oxidation (on catalyst surface or in gas phase) at high O₂ conversions. In particular, the ethane selectivity increased to ~15% at about 10% O₂ conversion, while the corresponding formaldehyde selectivity nearly declined to zero. Following this reviewer's suggestion, we have plotted the selectivity of methane oxidation on B₂O₃/Al₂O₃ catalysts as a function of O₂ conversion. As shown in Figure R4, when the O₂ conversion increased from 1.1% to 13.2% (corresponding to a change of methane conversion from 2.0% to 24.1%), the formaldehyde selectivity decreased from 61.0% to 8.3%, concomitant with the selectivity increases from 31.0% to 78.1% for CO and from 2.5% to 11.3% for CO₂. These results are consistent with the fact that formaldehyde is the primary product of methane oxidation, and CO and CO₂ are mainly formed from subsequent oxidation of formaldehyde. In contrast to the said reference, the summed selectivity of C₂ products (i.e. ethane and ethene) was below 3% even at a high O₂ conversion of 13.2%, which implies at least the concentration of methyl radical in gas phase is quite low under our study conditions and further supports the negligible role of gas-phase chemistry for methane oxidation on the B₂O₃-based catalysts.

Figure R4. Product selectivities as a function of O₂ conversion on the 20 wt% B₂O₃/Al₂O₃ catalyst. Reaction condition: 550°C, 32 kPa P_{CH₄}, 32 kPa P_{O₂}, N₂ used as the balance gas, WHSV= 2325-11625 mL/g_{cat}-h.

Response to Reviewer 2:

Reviewer 2: *The authors present their work on the conversion of methane-oxygen mixtures to formaldehyde and CO over B₂O₃-based catalyst at temperatures of 550°C and nearly atmospheric pressure. The catalyst is tested in a fixed bed reactor, the product stream analyzed by GC. High selectivity to HCHO and CO (ratio 1) at rather low methane conversion is achieved. Aside from experimental data, a mechanistic scheme is proposed.*

My major concern with this paper is the first sentence in the conclusion: “nonmetallic B₂O₃-based catalysts show superior selectivity and stability in selectively catalyzing the partial oxidation of methane to HCHO and CO.” Superior to what? Concerning CO, there are several catalysts that exhibit much higher selectivity and (even more) yields than the one proposed, just think of all the syngas production processes. Similar for HCHO, there are well-established commercial processes with excellent performance available. Since the paper as well as the journal focuses on catalysts, the authors should at least give a turn-over-frequency of their catalyst and a comparison to the established ones.

Response: Thank you for pointing out this issue. We totally agree with this reviewer that the first sentence in the conclusion is ambiguous and we have removed “superior” from it, which now becomes

“In summary, nonmetallic B₂O₃-based catalysts are selective and stable in the partial oxidation of methane to HCHO and CO.”

We have further followed this reviewer’s suggestion to include catalyst-weight-based reaction rates and turnover frequencies in Table S2 of the supporting information (attached below as Table R1) for the performance comparison of typical solid catalysts on partial oxidation of methane. It can be seen that the estimated turnover frequencies of B₂O₃/Al₂O₃ catalysts are an order of magnitude lower than those of V₂O₅-based catalysts that are among the most active ones, partially due to the lower reaction temperature conducted for the former. On the other hand, the B₂O₃/Al₂O₃ catalysts show much lower selectivities of the fully-oxidized CO₂ product compared with those reducible metal oxide catalysts, consistent with our hypothesis that the modest activity of the supported B₂O₃ catalysts renders selective oxidation of methane to partially-oxidized products (e.g. formaldehyde and CO).

Table R1. Comparison of activity and selectivity among typical solid catalysts for partial oxidation of methane.

Catalysts	Temp. (°C)	P _{CH₄} (kPa)	O ₂ /CH ₄ ratio	CH ₄ conversion (%)	Carbon selectivity (%)				Reaction rate (mmol _{CH₄} g _{cat} ⁻¹ h ⁻¹)	Turnover frequency (h ⁻¹)	Ref.
					HCHO	CH ₃ OH	CO	CO ₂			
MoO ₃ /SiO ₂	550	16.8	0.44	1.2	12.0	1.0	35.0	51.0	--	--	a
VO _x /SBA-15	600	20.2	0.13	1.8	36.4	0.8	58.7	4.5	97	126 ^k	b
CuO _x /SBA-15	625	33.8	1.00	2.8	44.0	--	28.0	28.0	17.5	--	c
FePO ₄ /SBA-15	650	33.8	0.50	3.3	36.0	--	--	--	7.3	21.9 ^k	d
VO _x /SiO ₂	650	--	0.11	4.5	32.0	--	57.0	11.0	88	220 ^k	e
VO _x /MCM-41	650	53.8	0.13	5.4	22.0	0.2	--	--	210	374 ^k	f
CsPW ₁₁ CoO ₃₉	650	72.1	0.40	6.0	22.5	6.6	34.9	36.0	--	--	g
WO _x /SiO ₂	650	84.9	0.19	6.9	11.9	--	25.1	63.0	7.8	12.0 ^k	h
MgO-B ₂ O ₃ /SiO ₂	600	38.0	0.33	3.5	18.6	--	--	--	1.3	0.7 ^k	i
MoO ₃ -SiO ₂	620	90.0	0.11	~4.8	~27		~48	~14	--	--	j
B ₂ O ₃ /Al ₂ O ₃	550	32.0	1.0	5.9	41.1	0.3	53.1	3.8	9.0	16.8 ^k	this work
B ₂ O ₃ /Al ₂ O ₂	550	32.0	2.0	6.8	46.0	0.1	50.4	2.7	10.4	19.4 ^k	this work

(a) *Catal. Today*, 1998, 45, 29-33; (b) *Appl. Catal. A*, 2003, 249, 345-354; (c) *J. Phys. Chem. C*, 2008, 112, 13700-13708; (d) *Catal. Today*, 2004, 93, 155-161; (e) *Top. Catal.*, 2017, 60, 1129-1139; (f) *J. Catal.* 2000, 191, 384-400; (g) *J. Mol. Catal. A*, 2013, 379, 255-262; (h) *Appl. Catal. A*, 1999, 184,143-152; (i) *J. Catal.*, 1987, 108, 252-255; (j): *AIChE J.*, 1987, 33, 1808-1812; (k) estimated assuming that all the sites exposed on the catalyst are active for methane oxidation.

REVIEWERS' COMMENTS

Reviewer #1 (Remarks to the Author):

I have read the response to my comments and I am happy that my concerns have now been answered mainly

I will still disagree about the gas phase nature of the reaction but I will not stand in teh way of publication.

Reviewer #1 (Remarks to the Author):

***Reviewer 1:** I have read the response to my comments and I am happy that my concerns have now been answered mainly. I will still disagree about the gas phase nature of the reaction but I will not stand in the way of publication.*

***Response:** We appreciate this reviewer's acknowledgement on our effort in elucidating the reaction mechanism and would like to thank this reviewer and the others again for their time and efforts for improving this manuscript.*